# Counterfactual Inference of Second Opinions

**Nina L. Corvelo Benz**[1,2]                    **Manuel Gomez Rodriguez**[1]

[1]Max Planck Institute for Software Systems, Kaiserslautern, Germany
[2]Department of Biosystems Science and Engineering, ETH Zurich, Zurich, Switzerland

## Abstract

Automated decision support systems that are able to infer second opinions from experts can potentially facilitate a more efficient allocation of resources—they can help decide when and from whom to seek a second opinion. In this paper, we look at the design of this type of support systems from the perspective of counterfactual inference. We focus on a multiclass classification setting and first show that, if experts make predictions on their own, the underlying causal mechanism generating their predictions needs to satisfy a desirable set invariant property. Further, we show that, for any causal mechanism satisfying this property, there exists an equivalent mechanism where the predictions by each expert are generated by independent sub-mechanisms governed by a common noise. This motivates the design of a set invariant Gumbel-Max structural causal model where the structure of the noise governing the sub-mechanisms underpinning the model depends on an intuitive notion of similarity between experts which can be estimated from data. Experiments on both synthetic and real data show that our model can be used to infer second opinions more accurately than its non-causal counterpart.

## 1 INTRODUCTION

In decision making under uncertainty, seeking opinions from multiple human experts tends to improve the overall quality of the decisions. For example, in medicine, second opinions have been shown valuable for establishing diagnoses and initiating treatment [Burger et al., 2020] as well as reducing the number of unnecessary procedures [Leape, 1989, Althabe et al., 2004]. In machine learning, ground truth labels are determined by carefully aggregating multiple noisy la-

bels provided by different experts [Zhang et al., 2016] and inconsistencies between these noisy labels help developing more robust models [Peterson et al., 2019]. Unfortunately, the timeliness and quality of the decisions is often compromised due to a shortage of experts, which prevents each decision to be informed by multiple experts' opinions.

In this context, we argue that the development of automated decision support systems that, given an expert's opinion on a decision instance and a set of features, are able to infer other experts' opinions will enable a more efficient allocation of resources. On the one hand, these systems could prevent (prioritize) seeking other experts' opinions when they are unlikely (likely) to bring new perspectives. On the other hand, these systems could also help identify those experts whose opinion is most likely to disagree with that of the expert sought first. Here, it is worth noting that several studies have also argued that decision support systems that identify disagreement between experts may help identify when a decision instance would benefit most from a second opinion [Raghu et al., 2019, Lim et al., 2021]. However, these studies do not focus on inferring other experts' opinions given an expert's opinion on a decision instance and a set of features, as we do in our work.

More specifically, we consider a multiclass classification setting where, for each instance, experts form their opinions on their own (*i.e.*, without communicating).[1] In this setting, each expert's opinion reduces to a label prediction. Then, our goal is to design decision support systems that, given an expert's prediction on an instance with a set of features, are able to infer other experts' predictions about the same instance, as illustrated in Figure 1. To this end, one could resort to standard supervised learning. Under this perspective, for each instance, the given expert's prediction would be just an additional feature about the instance. Unfortunately, this would limit the applicability of the resulting supervised learning model to the unrealistic scenario where, for each

---

[1]This setting fits a variety of real-world applications. For example, when a patient is diagnosed by multiple doctors separately.

*Accepted for the 38th Conference on Uncertainty in Artificial Intelligence* (UAI 2022).

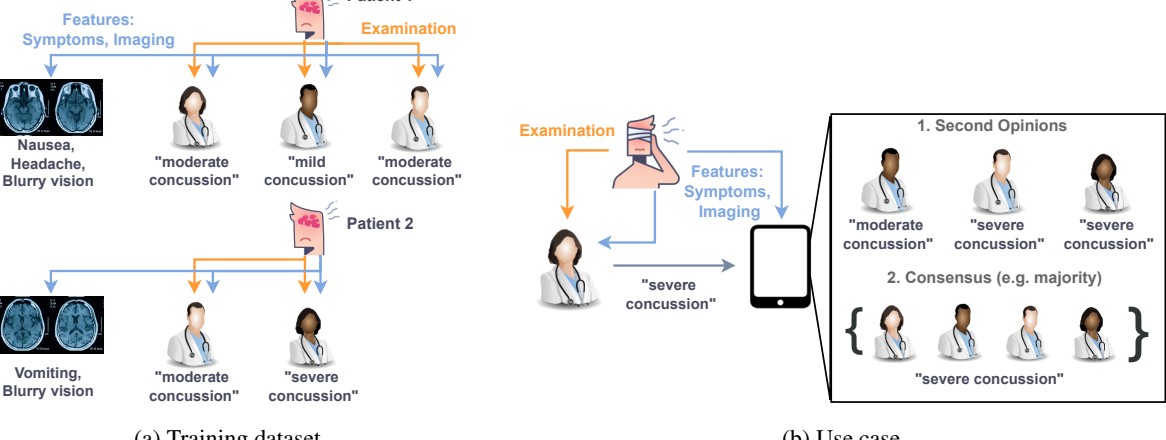

(a) Training dataset          (b) Use case

Figure 1: An example of a training dataset and use case of our decision support system on a medical application. In panel (a), for each patient, multiple doctors assess the severity of a concussion on the basis of a set of features. In panel (b), given a doctor's assessment of the severity of a concussion and a set of features, our decision support system infers other fellow doctors' assessment of the severity of the concussion.

possible pair of experts, we observe a sizeable number of instances where both experts made a prediction. To circumvent this limitation, we look at the design of the above systems from the perspective of counterfactual inference.

**Our contributions.** We first show that, if experts form their opinions for each instance on their own, the underlying causal mechanism generating the experts' predictions needs to satisfy a certain set invariant property. Moreover, we further show that any structural causal model satisfying the above set invariant property (in short, any SI-SCM) also satisfies two additional desirable properties:

(i) there exists an equivalent SI-SCM where each expert's predictions are generated by independent sub-mechanisms governed by a common (multidimensional) noise;

(ii) given an expert's prediction on an instance with a set of features, the conditional interventional distribution and the counterfactual distribution of another expert's predictions entailed by the SI-SCM are identical[2].

These properties suggest the following natural strategy to design and train SI-SCM based decision support systems. In a first step, we can use interventional data about each expert—her predictions on a set of instances—to determine the structure of each sub-mechanism separately. One can view this step as multiple independent supervised learning problems, one per expert. In a second step, we can use a small amount of interventional data about multiple experts making predictions about a joint set of instances to char-

acterize similarity across experts and factorize the noise governing the sub-mechanisms into a set of noise components. In a way, in this second step, we are adding a wrapper to the supervised learning models characterizing each expert's sub-mechanism to be able to make counterfactual predictions about second opinions.

To implement the above strategy, we introduce a specific class of SI-SCMs based on the Gumbel-Max structural causal model [Oberst and Sontag, 2019] (in short, Gumbel-Max SI-SCM) and characterize similarity across pairs of experts using the concept of counterfactual stability[3]. In the Gumbel-Max SI-SCM, each expert's sub-mechanism is governed by a Gumbel-Max noise variable and submechanisms of similar experts may be governed by the same noise variable. Further, we show that the problem of uniquely associating each of these noise variables with disjoint sets of mutually similar experts given data can be formulated as a known clique partioning problem, an NP-hard problem [Grötschel and Wakabayashi, 1989, 1990], and propose a randomized greedy algorithm with good performance.

Finally, we experiment with synthetic and real data comprising of $20,426$ expert predictions over $1,560$ natural images. The results on synthetic data show that our randomized greedy algorithm can successfully recover the disjoint sets of mutually similar experts underpinning a specific Gumbel-Max SI-SCM from data. The results on real data show that the (counterfactual) predictions provided by the Gumbel-Max SI-SCM are more accurate than those provided by its non-causal counterpart.

---

[2]Under the conditional interventional distribution, both experts have made a prediction but we only observe one of them. Under the counterfactual distribution, only one expert has made a prediction, which we observe.

[3]Counterfactual stability is, in general, an axiomatic requirement imposed to counterfactual distributions [Oberst and Sontag, 2019]. However, in SI-SCMs, it is verifiable from interventional data due to (ii), as shown in Theorem 7.

**Further related work.** Predictions by different experts have been typically studied separately, *i.e.*, without conditioning on an observed prediction by a given expert [Dawid and Skene, 1979, Welinder and Perona, 2010, Guan et al., 2018, Kerrigan et al., 2021, Straitouri et al., 2022]. One could think of the observed prediction just as an additional feature when inferring other experts' predictions, however, this would limit the applicability of existing inference methods to scenarios where, for each pair of experts, we observe a sizeable number of instances where both experts made a prediction, as discussed previously. More broadly, our work is not the first to use counterfactual reasoning in expert prediction [Bica et al., 2020]. However, previous work has used counterfactual reasoning to quantify an expert's preference over counterfactual outcomes rather than to infer other experts' predictions given an observed expert's prediction.

Counterfactual inference has a long and rich history [Imbens and Rubin, 2015]. However, it has mostly focused on estimating quantities related to the interventional distribution of interest such as, *e.g.*, the conditional average treatment effect (CATE). A few notable exceptions are by Oberst and Sontag [2019] and Tsirtsis et al. [2021], which use the Gumbel-Max SCM to reason about counterfactual distributions in Markov decision processes (MDPs), and by Lorberbom et al. [2021], which introduces a parameterized family of causal mechanisms that generalize the Gumbel-Max SCM and are specifically-tuned to a distribution of observations and interventions of interest. However, the Gumbel-Max structural causal model has not been used previously to reason about counterfactual expert predictions.

## 2 PRELIMINARIES

Given a set of random variables[4] $\mathbf{X} = \{X_1, \ldots X_n\}$, a structural causal model (SCM) $\mathcal{M}$ defines a complete data-generating process via a collection of assignments

$$X_i = f_i(\mathbf{PA}_i, U_i),$$

where $\mathbf{PA}_i \subseteq \mathbf{X} \setminus X_i$ are the direct causes of $X_i$, $\mathbf{F} = \{f_1, \ldots, f_n\}$ are deterministic causal mechanisms, $\mathbf{U} = \{U_1, \ldots, U_n\}$ are jointly independent noise random variables, and $P(\mathbf{U})$ denotes the (prior) distribution of the noise variables. Here, note that the noise variables $\mathbf{U}$ are the only source of stochasticity and, given an observational distribution $P(\mathbf{X})$, there always exists a distribution $P(\mathbf{U})$ and mechanisms $\mathbf{F}$ so that $P = P^{\mathcal{M}}$, where $P^{\mathcal{M}}$ is the distribution entailed by $\mathcal{M}$.

Two SCMs $\mathcal{M}$ and $\tilde{\mathcal{M}}$ over variables $\mathbf{X}$ and $\mathbf{U}$, with noise distribution $P(\mathbf{U})$ and mechanisms $\mathbf{F}$ and $\tilde{\mathbf{F}}$ respectively, are equivalent if, for all $i \in [n]$, it holds that

$$x_i = f_i(\mathbf{pa}_i, u_i) \iff x_i = \tilde{f}_i(\mathbf{pa}_i, u_i).$$

---

[4] We denote random variables with capital letters and realizations of random variables with lower case letters.

for any realization $\mathbf{PA}_i = \mathbf{pa}_i$ and $P(\mathbf{U})$-almost every $u_i$.[5]

Given a SCM $\mathcal{M}$, an atomic intervention $\mathcal{I}$ corresponds to assigning a fixed value to a variable. For example, let $\mathcal{I} = \text{do}[X_i = x]$ be the intervention that assigns value $x$ to variable $X_i$, then the intervened SCM $\mathcal{M}^{\mathcal{I}}$ does not assign the value of $X_i$ according to $f_i(\mathbf{PA}_i, U_i)$ but assign it to a fixed value $x$. The interventional distribution entailed by the intervened SCM is denoted $P^{\mathcal{M};\mathcal{I}}$. Furthermore, given the (possibly partial) observation $\mathbf{X} = \mathbf{x}$, we can also define a modified SCM $\mathcal{M}_{\mathbf{X}=\mathbf{x}}$ where the noise variables $\mathbf{U}$ are distributed according to the posterior distribution $P(\mathbf{U} \mid \mathbf{X} = \mathbf{x})$. Then, we can view a counterfactual statement as an intervention $\mathcal{I}$ in the SCM $\mathcal{M}_{\mathbf{X}=\mathbf{x}}$ and denote the counterfactual distribution entailed by the counterfactual SCM $\mathcal{M}^{\mathcal{I}}_{\mathbf{X}=\mathbf{x}}$ as $P^{\mathcal{M} \mid \mathbf{X}=\mathbf{x};\mathcal{I}}$.

The Gumbel-Max SCM is a specific class of SCM in which the causal mechanism for a random categorical variable $V$ is defined as

$$f_v(\mathbf{PA}, \mathbf{U}) := \underset{j}{\arg\max}\{\log P(V = j \mid \mathbf{PA}) + U_j\} \quad (1)$$

and each noise variable $U_j \sim \text{Gumbel}(0, 1)$. Here, note that the interventional distribution $P^{\mathcal{M}; do[\mathbf{PA}=\mathbf{pa}]}(V)$ entailed by a Gumbel-Max SCM $\mathcal{M}$ is exactly $P(V \mid \mathbf{PA} = \mathbf{pa})$.

## 3 COUNTERFACTUAL INFERENCE OF SECOND OPINIONS

We consider a multi-class classification task where, for each instance, a human expert $h \subseteq \mathcal{H}$ makes a label prediction $y_h \in \mathcal{Y} = \{1, \ldots, k\}$ based on multiple sources of information, which are (imperfectly) summarized by a feature vector $x \in \mathcal{X}$. Here, we assume that experts make predictions on their own (*i.e.*, without communicating with each other) and the assignment of experts to instances is independent of the identity of the instances and their feature vectors. Then, our goal is to design an automated decision support system that, given a prediction $y_h$ from an expert $h$ about an instance summarized by a feature vector $x$, is able to infer what prediction $y_{h'}$ another expert $h' \neq h$ would have made about the *same* instance if she had been asked. Here, note that two different instances may be (imperfectly) summarized by the same feature vector $x$, however, we are interested in a counterfactual prediction about the *same* instance.

Our starting point is to view the above counterfactual statement as an intervention in a particular counterfactual SCM. More specifically, let $\mathcal{M}$ be a SCM defined by the assignments

$$\mathbf{Y} = f_{\mathbf{Y}}(X, Z, U), \quad Z = f_Z(V), \quad \text{and} \quad X = f_X(W) \quad (2)$$

---

[5] $P(\mathbf{U})$-almost everywhere means that the set of noise realizations $\mathcal{U}'$ for which the property does not hold has probability zero under the distribution $P(\mathbf{U})$, *i.e.*, $P(\mathbf{U} \in \mathcal{U}') = 0$.

where $U$, $V$ and $W$ are (multidimensional) independent noise variables, $f_{\mathbf{Y}}$, $f_Z$ and $f_X$ are given deterministic causal mechanisms (or functions), and $\mathbf{Y} = (Y_h)_{h \in Z}$ are the predictions by a set of human experts $Z \subseteq \mathcal{H}$. Then, we can express the above counterfactual statement as an intervention $\mathcal{I} = do[Z = \{h'\}]$ in the counterfactual SCM $\mathcal{M}_{X=x, Z=\{h\}, \mathbf{Y}=y_h}$ and, to infer the label prediction $y_{h'}$, we just need to resort to the counterfactual distribution $P^{\mathcal{M} \mid X=x, Z=\{h\}, \mathbf{Y}=y_h \,; do[Z=\{h'\}]}(\mathbf{Y})$.

At this point, one may argue that, even if we find a noise distribution $P(U)$ and a function $f_{\mathbf{Y}}$ under which the conditional distribution $P^{\mathcal{M}}(\mathbf{Y} \mid X)$ is a good fit for observed historical predictions by experts, we would be unable to validate how accurate our counterfactual label predictions are using data. In general, this is true since counterfactual reasoning lies within level three in the "ladder of causation" [Pearl, 2009]. In this context, previous work resorts instead to axiomatic assumptions about the causal mechanism of the world [Oberst and Sontag, 2019, Tsirtsis et al., 2021, Noorbakhsh and Rodriguez, 2021]. In our setting, this would reduce to specifying how differences across experts may have lead to a different prediction while holding "everything else" fixed. However, in what follows, we will show that, if experts do not communicate with each other, the above SCM satisfies a set invariance property that surprisingly implies that the above counterfactual distribution coincides with an interventional conditional distribution. This enables a data-driven design and validation of our SCM based decision support system.

## 4   RELATING THE COUNTERFACTUAL AND INTERVENTIONAL WORLDS

To build some intuition on the reasons why, if experts do not communicate, certain type of counterfactual and interventional distributions are identical, we start with a simple example. Let $h, h' \in \mathcal{H}$ be two different experts and consider the following two questions:

1. Both experts have made a label prediction about an instance (*i.e.*, $Z = \{h, h'\}$) but we only observe the prediction $Y_h = c$ made by $h$, what is the prediction made by $h'$?

2. One of the experts has made a label prediction $Y_h = c$ about an instance (*i.e.*, $Z = \{h\}$) and we observe it, what would the prediction made by $h'$ be if she had made a prediction?

The first question is of conditional nature while the second is a counterfactual one. In general, the answer to both questions may differ, for example, if experts influence each other's predictions by sharing and discussing their opinions in the first case. However, if experts do not communicate, the answer to both questions should be identical. More formally, the following conditional interventional distribution

and counterfactual distribution of the expert should be equal:

$$P^{\mathcal{M} \,; do[Z=\{h,h'\}]}(Y_{h'} \mid X = x, Y_h = c)$$
$$= P^{\mathcal{M} \mid X=x, Z=\{h\}, \mathbf{Y}=c \,; do[Z=\{h'\}]}(\mathbf{Y}). \quad (3)$$

More generally, we will now show that, if each expert forms their opinion on their own, the above equality is a direct consequence of a set invariance property satisfied by the SCM defined in Eq. (2).

**Set Invariant SCMs (SI-SCMs).** If experts do not communicate before making a prediction and hence are unaware and unaffected of other experts' opinions, the mechanism $f_{\mathbf{Y}}$ has a set invariant value over expansions (supersets) of $Z$. For example, consider one single expert $h$ has made a prediction $f_{\mathbf{Y}}(x, \{h\}, u) = c$ about a specific instance. Then, one can conclude that, if instead of a single expert, a set of experts $\zeta \subseteq \mathcal{H}$ such that $h \in \zeta$ would have made predictions about the same instance (*i.e.*, $x$ and $u$ does not change), expert $h$ would have made the same prediction, *i.e.*, $(f_{\mathbf{Y}}(x, \zeta, u))_h = c$. More formally, we define the set invariance property as follows:

**Definition 1** (Set Invariance). *A mechanism $f_{\mathbf{Y}}$ for variable $\mathbf{Y}$ is set invariant with respect to $Z$ if, for any two realizations $Z = \zeta$ and $Z = \zeta'$ such that $\zeta \subseteq \zeta'$, it holds that*

$$f_{\mathbf{Y}}(x, \zeta, u) = (f_{\mathbf{Y}}(x, \zeta', u))_\zeta \quad \text{for all } x \in \mathcal{X}, u \in \mathcal{U}.$$

*A SCM $\mathcal{M}$ with such a mechanism is set invariant for $\mathbf{Y}$.*

A set-invariant SCM (SI-SCM) for $Y$ can be constructed by expressing the causal mechanism $f_{\mathbf{Y}}$ with sub-mechanisms $f_{Y_h}$ governed by a common noise variable:[6]

**Theorem 2.** *Any SCM $\mathcal{M}$ with mechanism $f_{\mathbf{Y}}$ of the form $f_{\mathbf{Y}}(X, Z, U) = (f_{Y_h}(X, U))_{h \in Z}$, where $f_{Y_h} : \mathcal{X} \times \mathcal{U} \to \mathcal{Y}$ are arbitrary functions, is set invariant for $\mathbf{Y}$.*

In fact, the following theorem shows that the class of SCMs with separate sub-mechanisms for $Y_h$ and a shared noise variable $U$ is not only a subclass but completely defines the class of SI-SCMs for $\mathbf{Y}$. Thus, any correlation between experts' predictions is caused by the common noise and features but not the causal mechanism.

**Theorem 3.** *For any SI-SCM $\mathcal{M}$, there exists an equivalent SI-SCM $\mathcal{M}'$ with causal mechanism $f'_{\mathbf{Y}}(X, Z, U) = (f'_{Y_h}(X, U))_{h \in Z}$ where for $h \in Z$*

$$f'_{Y_h}(X, U) := (f_{\mathbf{Y}}(X, \{h\}, U))_{h \in Z}.$$

Here, we would like to emphasize that, if the mechanism $f_{\mathbf{Y}}$ of an SCM is not explicitly decoupled into sub-mechanisms governed by the same noise, it may be challenging to check

---

[6]All proofs can be found in Appendix 1

whether an arbitrary SCM is set invariant. For arbitrary SCMs, Theorem 2 can not be applied directly and Theorem 3 does not tell us how to verify that an equivalent SCM exists. However, it tells us that the mechanism of a set invariant SCM can be decoupled and simplified. It would be interesting to develop methods to check for set invariance for arbitrary SCMs in future work.

**Equality between the counterfactual distribution and the conditional interventional distribution.** Returning to our simple motivational example, note that, if a SCM is set invariant, the answers to the counterfactual and the conditional questions 1 and 2 are the same as long as the noise $u \sim P(U \mid X = x, Y_h = c)$ is the same. In particular, for question 1, the answer is $Y_{h'} = (f_{\mathbf{Y}}(x, \{h, h'\}, u))_{h'}$, for question 2, the answer is $Y_{h'} = f_{\mathbf{Y}}(x, \{h'\}, u)$, and since $f_{\mathbf{Y}}$ is set invariant, both answers are equal. More generally, for arbitrary sets of experts, we can easily conclude that equality holds if and only if $f_{\mathbf{Y}}$ is set invariant.

Next, to show that, if a SCM is set invariant, then the equality of distributions in Eq. (3) holds, we first present a more general theorem that states that, if we expand the set of experts who make predictions, the corresponding interventional distribution of $\mathbf{Y}$ does not change:

**Theorem 4.** *Let SCM $\mathcal{M}$ be set invariant for $\mathbf{Y}$. Then, for any $\zeta, \zeta' \in \mathcal{H}$ such that $\zeta \subseteq \zeta'$, it holds that*

$$
P^{\mathcal{M}\,;\,do[Z=\zeta]}(\mathbf{Y} = \mathbf{y} \mid X) \\
= P^{\mathcal{M}\,;\,do[Z=\zeta']}((\mathbf{Y})_\zeta = \mathbf{y} \mid X)
$$

*for any $\mathbf{y} \in \mathcal{Y}^{|\zeta|}$ where $(\mathbf{Y})_\zeta$ denotes the predictions by the experts in the subset $\zeta \subseteq \zeta'$.*

The above theorem is straight forward to show using that, due to the set invariance property, the prediction values of mechanism $f_{\mathbf{Y}}$ for $(x, \zeta, u)$ are equal to the values for $(x, \zeta', u)$ for experts in $\zeta$ and, due to the independence between the noise and the intervention, the noise distribution does not change. A direct conclusion is that, no matter how many experts make predictions, the conditional interventional distribution of a single expert's prediction does not change, as formalized by the following corollary:

**Corollary 1.** *Let SCM $\mathcal{M}$ be set invariant for $\mathbf{Y}$. Then, for any $h \in \mathcal{H}$ and $\zeta \subseteq \mathcal{H}$ such that $h \in \zeta$, it holds that*

$$
P^{M\,;\,do[Z=\{h\}]}(Y_h \mid X) = P^{M\,;\,do[Z=\zeta]}(Y_h \mid X).
$$

Similarly, we can derive the desired equality between the counterfactual distribution and the conditional interventional distribution by using the set invariance of mechanism $f_{\mathbf{Y}}$ and the fact that the noise distribution changes equally in both scenarios. More formally, we have the following corollary:

**Corollary 2.** *Let SCM $\mathcal{M}$ be set invariant for $\mathbf{Y}$. Then, for any $h, h' \in \mathcal{H}$ and $\zeta \subseteq \mathcal{H}$ such that $h, h' \in \zeta$, it holds that*

$$
P^{\mathcal{M}\,|\,X=x, Z=\{h\}, \mathbf{Y}=c\,;\,do[Z=\{h'\}]}(\mathbf{Y}) \\
= P^{\mathcal{M}\,;\,do[Z=\zeta]}(Y_{h'} \mid X = x, Y_h = c).
$$

*for any $x \in \mathcal{X}$ and $c \in \mathcal{Y}$.*

**Remark.** While we have introduced the notion of set invariance for SCMs in the context of inferring second opinions, we believe it may be of independent interest since, generally speaking, it allows us to identify counterfactual distributions from interventional data.

# 5 CHARACTERIZING MUTUALLY SIMILAR EXPERTS

Given a SI-SCM model $\mathcal{M}$ where each expert's predictions $Y_h$ are generated by a sub-mechanism $f_{Y_h}$, our goal in this section is to characterize mutually similar experts. Later on, this will help us factorize the noise $U$ governing the sub-mechanisms $f_{Y_h}$ underpinning the model into a set of independent noise components and uniquely associate each of these noise components with disjoint sets of mutually similar experts given data.

To this end, we first start by characterizing similarity between a pair of experts $h, h' \in \mathcal{H}$. To this end, we resort to the recently introduced notion of counterfactual stability [Oberst and Sontag, 2019]. More specifically, we argue that two experts $h$ and $h'$ are *similar* if $\mathcal{M}$ satisfies counterfactual stability for $h, h'$ with respect to $\mathbf{Y}$.

**Definition 5** (Counterfactual stability)**.** *A SCM $\mathcal{M}$ satisfies counterfactual stability for $h, h'$ with respect to $\mathbf{Y}$ if, for all $\zeta, \zeta' \subseteq \mathcal{H}$ such that $h \in \zeta$ and $h' \in \zeta'$ and for all $c' \neq c$, the condition*

$$
\frac{P^{\mathcal{M}\,;\,do[Z=\zeta']}(Y_{h'} = c \mid X)}{P^{\mathcal{M}\,;\,do[Z=\zeta]}(Y_h = c \mid X)} \geq \frac{P^{\mathcal{M}\,;\,do[Z=\zeta']}(Y_{h'} = c' \mid X)}{P^{\mathcal{M}\,;\,do[Z=\zeta]}(Y_h = c' \mid X)}
$$

*implies that $P^{\mathcal{M}\,|\,X, Z=\zeta, Y_h=c\,;\,do[Z=\zeta']}(Y_{h'} = c') = 0$, where $Y_h = c$ is the observed outcome under $do[Z = \zeta]$.*

For example, consider a scenario where a doctor needs to decide what treatment option—surgery ($Y = 0$), radiation ($Y = 1$) or chemotherapy ($Y = 2$)—will be more beneficial for a patient with a tumor, imperfectly summarized by a feature vector $x$. Assume doctor $h$ decides the most beneficial option is surgery, *i.e.*, $Y_h = 0$, and we know that, for patients with similar $x$, doctor $h'$ is generally more likely to operate and less likely to resort to therapy than doctor $h$. Then, if doctors $h$ and $h'$ are similar, as defined in Definition 5, we expect doctor $h'$ would have also decided the most beneficial option is surgery for the given patient, if consulted, *i.e.*, $Y_{h'} = 0$. Here, whenever two doctors $h$ and

$h'$ are *not* similar, one could argue that it is because they weigh any (hidden) factor of the patient at hand differently[7].

Unfortunately, in general, we cannot use data to verify if two experts $h$ and $h'$ are similar. This is because our notion of similarity relies on a counterfactual distribution, $P^{\mathcal{M} \mid Y_h = c \,;\, \mathrm{do}[Z = \zeta']}$, and counterfactual reasoning lies within level three in the "ladder of causation" [Pearl, 2009]. However, we will now define a notion of conditional stability that is verifiable using interventional data and, in the case of SI-SCMs, is both a sufficient and necessary condition for counterfactual stability—if conditional stability holds, we can conclude that two experts are similar.

**Definition 6** (Conditional stability). *A SCM $\mathcal{M}$ satisfies conditional stability for two experts $h, h' \in \mathcal{H}$ with respect to $\mathbf{Y}$ if, for all $\zeta \subseteq \mathcal{H}$ such that $h, h' \in \zeta$ and for all $c' \neq c$, the condition*

$$\frac{P^{\mathcal{M};do[Z=\zeta]}(Y_{h'} = c \mid X)}{P^{\mathcal{M};do[Z=\zeta]}(Y_h = c \mid X)} \geq \frac{P^{\mathcal{M};do[Z=\zeta]}(Y_{h'} = c' \mid X)}{P^{\mathcal{M};do[Z=\zeta]}(Y_h = c' \mid X)} \quad (4)$$

*implies that $P^{\mathcal{M};do[Z=\zeta]}(Y_{h'} = c' \mid X, Y_h = c) = 0$.*

Here, note that, for SI-SCMs, we only need to verify the condition in Eq. (4) for the sets $\zeta = \{h\}$ and $\zeta = \{h'\}$ because no matter how many experts make predictions, the conditional interventional distributions in Eq. (4) do not change, as shown in Corollary 1. Then, the following Theorem formalizes the equivalence between conditional and counterfactual stability:

**Theorem 7.** *Let SCM $\mathcal{M}$ be set invariant for $\mathbf{Y}$. Then, $\mathcal{M}$ satisfies counterfactual stability for $h, h' \in \mathcal{H}$ with respect to $\mathbf{Y}$ iff it satisfies conditional stability.*

Once we have a notion of similarity between pairs of experts that we can verify from data, we can characterize groups of mutually similar experts. In this context, it will be useful to introduce the following notion of pairwise counterfactual stability (in short, PCS), which extends counterfactual stability to groups of experts $\zeta \subseteq \mathcal{H}$ of arbitrary size.

**Definition 8** (Pairwise Counterfactual Stability). *A SCM $\mathcal{M}$ satisfies pairwise counterfactual stability for a group of experts $\zeta \subseteq \mathcal{H}$ with respect to $\mathbf{Y}$ if it satisfies counterfactual stability for any $h, h' \in \zeta$.*

Similarly as in the case with a pair of experts, one can also define pairwise conditional stability and it immediately follows from Theorem 7 that, for SI-SCM, pairwise conditional and counterfactual stability are equivalent, as formalized by the following Corollary.

---

[7] In general, note that similarity between experts does not always deterministically enforce the observed expert's prediction on the counterfactual prediction. In the example above, this happens because the inequality in Def. 5 holds for the two remaining label values. Rather, it allows us to identify experts with different decision making criteria.

**Corollary 3.** *Let SCM $\mathcal{M}$ be set invariant for $\mathbf{Y}$. Then, $\mathcal{M}$ satisfies pairwise counterfactual stability for $\zeta \in \mathcal{H}$ with respect to $\mathbf{Y}$ iff it satisfies pairwise conditional stability.*

# 6 GUMBEL-MAX SI-SCM

In this section, we build upon our theoretical results to develop the Gumbel-Max SI-SCM, a new class of SI-SCM based on the Gumbel-Max SCM.

Given a set of experts $\mathcal{H}$, the Gumbel-Max SI-SCM partitions $\mathcal{H}$ into disjoint sets of experts $\Psi = \{\psi\}_{\psi \in \Psi}$, as defined in Section 5, and associate all experts within each set to the same multidimensional noise variable. More formally, the Gumbel-Max SI-SCM is defined as follows:

**Definition 9** (Gumbel-Max SI-SCM). *The Gumbel-Max SI-SCM $\mathcal{M}(\Psi)$ is a specific class of SCM in which the causal mechanism for $\mathbf{Y}$ is defined as*

$$f_{\mathbf{Y}}(X, Z, U) = (f_{Y_h}(X, U))_{h \in Z}, \quad with$$

$$f_{Y_h}(X, U_{\psi(h)}) = \underset{c \in \mathcal{Y}}{\mathrm{argmax}}\{\log P(Y_h = c \mid X) + U_{\psi(h),c}\},$$

*where $\psi(h) \in \Psi$ denotes the subgroup expert $h$ belongs to and each noise variable $U_{\psi(h),c} \sim Gumbel(0, 1)$.*

By definition, the Gumbel-Max SI-SCM $\mathcal{M}(\Psi)$ is set invariant for $\mathcal{Y}$ and, for any $\zeta \subseteq \mathcal{H}$ and $h \in \zeta$, it holds that $P^{\mathcal{M}(\Psi);do[Z=\zeta]}(Y_h \mid X) = P(Y_h \mid X)$. Moreover, all experts within each group $\psi \in \Psi$ are mutually similar, as formalized by the following Theorem:

**Theorem 10.** *The Gumbel-Max SI-SCM $\mathcal{M}(\Psi)$ satisfies pairwise counterfactual stability (PCS) for each group $\psi \in \Psi$ with respect to $\mathbf{Y}$.*

Finally, note that, for $\Psi = \mathcal{H}$, the Gumbel-Max SI-SCM reduces to the original Gumbel-Max SCM defined in Eq. 1. Therefore, one can view the Gumbel-Max SI-SCM as a generalization of the original Gumbel-Max SCM where, instead of a single multidimensional noise variable $U$ for all $h \in \mathcal{H}$, one has several noise variables $U_\psi$, one per group.

**Estimating counterfactual distributions.** Given a prediction $Y_h = c$ by an expert $h$, we can compute an unbiased finite sample Monte-Carlo estimator of the counterfactual distribution $P^{\mathcal{M}(\Psi) \mid X=x, Z=\{h\}, \mathbf{Y}=y_h \,;\, do[Z=\{h'\}]}(\mathbf{Y})$ for the prediction $Y_{h'}$ of another expert $h' \neq h$ as follows:

$$P^{\mathcal{M}(\Psi) \mid X=x, Z=\{h\}, \mathbf{Y}=y_h \,;\, do[Z=\{h'\}]}(\mathbf{Y})$$
$$\approx \frac{1}{T} \sum_{t \in T} \mathbb{1}[c = f_{Y_{h'}}(x, \mathbf{u}_t)] \quad (5)$$

where $\mathbf{u}_1, \ldots, \mathbf{u}_T$ are samples from the posterior distribution $P^{\mathcal{M}(\Psi) \mid X=x, Z=\{h\}, \mathbf{Y}=y_h \,;\, do[Z=\{h'\}]}(U_{\psi(h')})$ of the

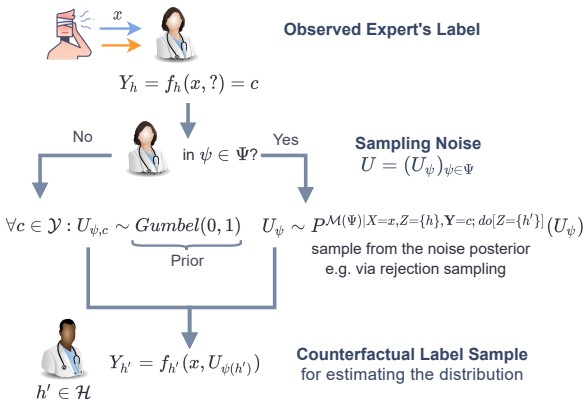

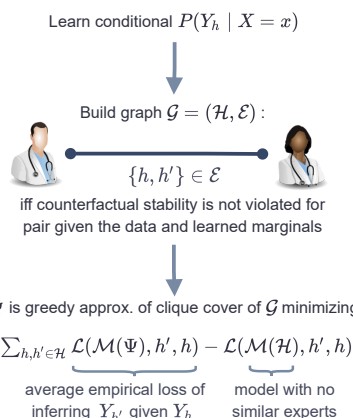

Figure 2: Illustration of the counterfactual sampling of experts' predictions with the Gumbel-Max SI-SCM $\mathcal{M}(\Psi)$.

Figure 3: Partitioning experts into mutually similar groups $\Psi$.

noise variable $U_{\psi(h')}$. Here, we can use an efficient procedure to sample from the above noise posterior distribution, described elsewhere [Oberst and Sontag, 2019, Maddison et al., 2015]. Moreover, note that, if $h \notin \psi(h')$, the posterior distribution coincides with the prior $P^{\mathcal{M}(\Psi)}(U_{\psi(h')})$. We summarized the sampling procedure is depicted in Figure 2.

**Partitioning experts into mutually similar groups.** In the Gumbel-Max SI-SCM $\mathcal{M}(\Psi)$, for each expert $h \in \mathcal{H}$, we can estimate the conditional distribution $P(Y_h \mid X)$ using any machine learning model trained using historical predictions made by the expert $h$. However, to fully define $\mathcal{M}(\Psi)$, we need to partition the set of experts $\mathcal{H}$ into disjoint sets of experts $\Psi$ given a small amount of historical data about multiple experts making predictions about a joint set of instances. To this end, we proceed as follows.

First, we look for violations of the conditional stability condition throughout the historical data. Whenever there exists a sample for which the predictions by two different experts $h$ and $h'$ violate conditional stability[8], we conclude that $h$ and $h'$ cannot belong to the same group $\psi$. Further, we also conclude that any pair of experts whose predictions did not violate conditional stability and were at least once observed for the same sample *can* be similar. However, since conditional stability is not a transitive property, there may be multiple valid partitions $\mathcal{P} = \{\Psi\}$ of the experts into disjoint sets that are consistent with the above conclusions. To decide among them, we would like to pick the partition $\Psi \in \mathcal{P}$ under which the counterfactual distributions $P^{\mathcal{M}(\Psi) \mid X=x, Z=\{h\}, \mathbf{Y}=y_h ; do[Z=\{h'\}]}(\mathbf{Y})$ provide the best goodness of fit. More formally, we would like to solve the following minimization problem:

$$\underset{\Psi \in \mathcal{P}}{\text{minimize}} \quad \sum_{h, h' \in \mathcal{H}} \mathcal{L}(\mathcal{M}(\Psi), h', h) - \mathcal{L}(\mathcal{M}(\mathcal{H}), h', h),$$
$$(6)$$

[8]A violation occurs whenever Eq. (4) holds but we observe $Y_h = c$ and $Y_{h'} = c'$.

where $\mathcal{L}(\mathcal{M}(\cdot), h', h)$ denotes an average (empirical) loss whenever we observe $Y_h$ and infer the label prediction $Y_{h'}$ using the counterfactual distribution $P^{\mathcal{M}(\cdot) \mid X=x, Z=\{h\}, \mathbf{Y}=y_h ; do[Z=\{h'\}]}(\mathbf{Y})$. Here, we measure goodness of fit in terms of average loss reduction with respect to the counterfactual distributions entailed by the causal model $\mathcal{M}(\mathcal{H})$ because this will allow us to reduce the number of pairs $(h, h')$ we need to consider. The step-wise approach for obtaining $\Psi$ is summarized in Figure 3.

Next, we formulate the above problem as a known clique partitioning problem [Grötschel and Wakabayashi, 1989, 1990]. More specifically, let $\mathcal{G} = (\mathcal{H}, \mathcal{E})$ be an undirected graph where, if $\{h, h'\} \in \mathcal{E}$, then $h$ and $h'$ *can* be similar, as concluded from the data. Then, it readily follows that finding a partition $\Psi$ of $\mathcal{H}$ is equivalent to finding a clique cover for $\mathcal{G}$[9]. Now, let the weight $w(h, h')$ of each edge $\{h, h'\} \in \mathcal{E}$ be given by:

$$w(h, h') = \mathcal{L}(\mathcal{M}(\Psi), h, h') - \mathcal{L}(\mathcal{M}(\mathcal{H}), h, h') + \mathcal{L}(\mathcal{M}(\Psi), h', h) - \mathcal{L}(\mathcal{M}(\mathcal{H}), h', h)$$

Then, we can rewrite the minimization problem in Eq. 6 as:

$$\underset{\Psi}{\text{minimize}} \quad \sum_{\psi \in \Psi} \sum_{h, h' \in \psi} w(h, h')$$
$$\text{subject to} \quad \Psi \text{ is a clique cover for } \mathcal{G}, \tag{7}$$

where note that we only need to consider pairs of experts $h, h' \in \psi$ because, otherwise, $w(h, h') = 0$ since the corresponding counterfactual distributions entailed by $\mathcal{M}(\Psi)$ and $\mathcal{M}(\mathcal{H})$ coincide. The minimization problem given by Eq. (7) is a known clique partitioning problem (CPP)[10], for which the decision problem of CPP for arbitrary weights is NP-Hard [Grötschel and Wakabayashi, 1989, 1990]. However,

[9]$\Psi$ is a clique cover for $\mathcal{G}$ iff $\Psi$ is a partition of $\mathcal{H}$, *i.e.*, $\bigcup_{\psi \in \Psi} \psi = \mathcal{H}$ and $\psi \cap \psi' = \emptyset$ for all $\psi, \psi' \in \Psi$, and vertices in $\psi \in \Psi$ form a clique in $\mathcal{G}$.

[10]In most of the literature, the problem is defined for complete

we found that a simple randomized greedy algorithm works well in our setting, as shown in Figure 1 in Appendix 3. Refer to Appendix 2 for more details about the algorithm.

# 7 EXPERIMENTS ON REAL DATA

In this section, we compare the performance of the proposed Gumbel-Max SI-SCM at inferring second opinions against several competitive baselines using a dataset with real expert predictions over natural images. Appendix 3 contains additional experiments on synthetic data where we assess the performance of Algorithm 1 at recovering groups of mutually similar experts on synthetic data.[11]

**Data description and experimental setup.** We experiment with the dataset CIFAR-10H [Peterson et al., 2019], which contains 10,000 images taken from the test set of the standard dataset CIFAR-10 [Krizhevsky et al., 2009]. Each of these images belongs to $n = 10$ classes and contains label predictions from approximately 50 human annotators. In total, the images are annotated by 2,571 different human annotators (from now on, experts).[12] Since the classification task is relatively easy for humans, there are many images (∼35%) in which there is full agreement between experts—all experts make the same label prediction. Here, motivated by the empirical observation that, in medical diagnosis, there is typically a 20% per-instance disagreement among experts [Van Such et al., 2017, Elmore et al., 2015], we filter out the above mentioned images in which there is full agreement. Moreover, we split the remaining images into two disjoint sets at random—a training set and a test set—and filter out data from any expert who made less than 130 and 20 predictions in training and test set, respectively, and whose predicted labels in the training data do not cover all class labels. After these preprocessing steps, the resulting training and test sets contain 1,257 and 303 images, respectively, annotated by $|\mathcal{H}| = 114$ experts, where each image in the training and test set is annotated by at least two experts.

To find the groups of mutually similar experts underpinning our Gumbel-Max SI-SCM, we run Algorithm 1 on the training set. Within the Gumbel-Max SI-SCM, we estimate the conditional distribution $P^{\mathcal{M};\mathrm{do}[Z=\{h\}]}(Y_h \mid X)$ for each expert $h$ using a Gaussian Naive Bayes model (GNB) trained

Table 1: Overall test accuracy

| Model | $h, h' \in \mathcal{H}$ | $h, h' \in \psi$ | $h \in \psi, h \in \psi'$ |
|---|---|---|---|
| Gumbel-Max SI-SCM | 66.8% | 79.9% | 45.1% |
| GNB | 48.9% | 51.3% | 45.1% |
| GNB + CNB | 62.0% | 66.0% | 55.2% |

using also the training set (one GNB per expert)[13]. Each GNB model uses 20 dimensional feature vectors computed by running PCA on a 512 dimensional normalized feature vector extracted using VGG19 [Simonyan and Zisserman, 2014]. Both during training and test, given an observed label prediction $Y_h$ by an expert $h$, we infer the prediction $Y_{h'}$ by another expert $h'$ using the most likely label under (an estimate of) the corresponding counterfactual distribution. To estimate each counterfactual distribution, we use $T = 1,000$ samples from the noise posterior distribution.

**Baselines and evaluation metrics.** We compare the performance of our trained Gumbel-Max SI-SCM with two baselines (see also Figure 2 in Appendix 4):

— The "*GNB*" baseline uses only the same Gaussian Naive Bayes models (GNB), one per expert, used by our trained Gumbel-Max SI-SCM. More specifically, given an observed label prediction $Y_h$ by an expert $h$, it infers the prediction $Y_{h'}$ by another expert $h'$ using the most likely label under the estimate of the conditional distribution $P^{\mathcal{M};\mathrm{do}[Z=\{h'\}]}(Y_{h'} \mid X)$ given by the corresponding GNB.

— The "*GNB + CNB*" baseline uses the same Gaussian Naive Bayes models (GNB), one per expert, used by our trained Gumbel-Max SI-SCM and a Categorical Naive Bayes (CNB) model, one per expert, that estimates $P^{\mathcal{M};\mathrm{do}[Z=\{h,h'\}]}(Y_{h'} \mid Y_h)$.[14] More specifically, given an observed label prediction $Y_h$ by an expert $h$, it infers the prediction $Y_{h'}$ by another expert $h'$ using the most likely label under the product of distributions $P^{\mathcal{M};\mathrm{do}[Z=\{h'\}]}(Y_{h'} \mid X) \times P^{\mathcal{M};\mathrm{do}[Z=\{h,h'\}]}(Y_{h'} \mid Y_h)$, as estimated by the corresponding GNB (first term) and CNB (second term).

To compare the performance of our trained Gumbel-Max SI-SCM and both baselines, for each sample in the test set, we pick each of the corresponding expert label predictions $Y_h$ as the observed prediction in turn and infer the value of the other predictions $Y_{h'}$. Here, we compute the overall

---

graphs. However, for arbitrary graphs, one can simply include the missing edges and assign positive infinite weights so that they are not included in a solution [Brimberg et al., 2017]

[11]To facilitate research in this area, we release an open-source implementation of our code at https://github.com/Networks-Learning/cfact-inference-second-opinions.

[12]The dataset CIFAR-10H is one of the only larger public datasets containing multiple label predictions by different experts per sample, necessary to train the proposed Gumbel-Max SI-SCM. However, since our methodology and theoretical results are rather general, our model may also be useful in other applications.

[13]In the CIFAR-10H dataset, experts are assigned to images (presumably) at random. Therefore, it holds that $P^{\mathcal{M};\mathrm{do}[Z=\{h\}]}(Y_h \mid X) = P(Y_h \mid X, Z = \{h\})$ and we can use observational data to estimate the interventional conditional distribution $P^{\mathcal{M};\mathrm{do}[Z=\{h\}]}(Y_h \mid X)$.

[14]The CNB uses a "one-hot" encoding of the observed prediction $Y_h$ as a single $|\mathcal{H}|$-dimensional feature where, for each dimension, it uses an additional label value to denote than an expert's label prediction has not been observed.

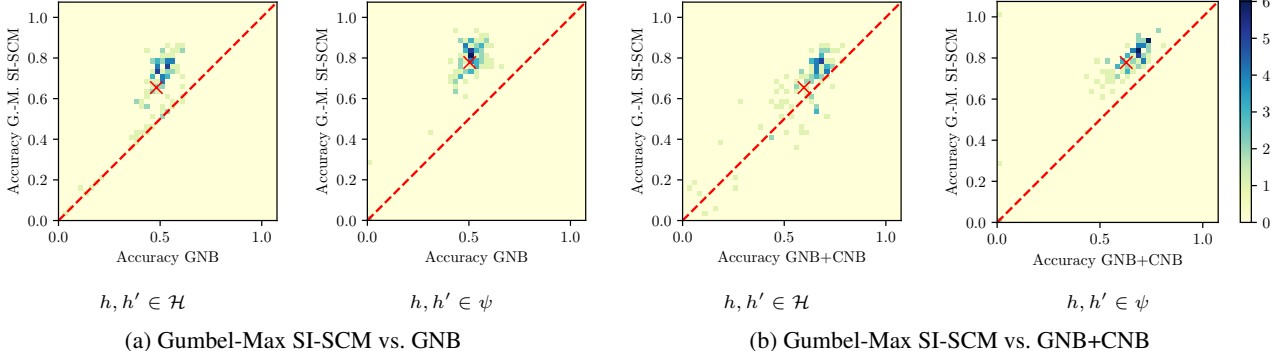

| $h, h' \in \mathcal{H}$ | $h, h' \in \psi$ | $h, h' \in \mathcal{H}$ | $h, h' \in \psi$ |
| :---: | :---: | :---: | :---: |
| (a) Gumbel-Max SI-SCM vs. GNB | | (b) Gumbel-Max SI-SCM vs. GNB+CNB | |

Figure 4: Per-expert test accuracy achieved by our model and both baselines on the CIFAR-10H dataset. In each panel, the $y$-axis measures the per-expert test accuracy achieved by our method and the $x$-axis the per-expert accuracy achieved by a baseline. For each cell, the darkness is proportional to the number of experts with the corresponding test accuracies.

accuracy as well as the per-expert accuracy and distinguish among three scenarios: (i) $h, h' \in \mathcal{H}$; (ii) $h, h' \in \psi$; and, (iii) $h \in \psi, h' \in \psi', \psi \neq \psi'$.

**Results.** We start by reporting that, during the training of our Gumbel-Max SI-SCM, Algorithm 1 found 352 violations of the conditional stability condition between pairs of experts and partitioned the experts into fifteen disjoint groups of mutually similar experts, where seven of these groups were singletons. Refer to Appendix 4 for more details regarding the groups identified by Algorithm 1.

Next, we report the overall accuracy achieved by our model and the baselines in Table 1. We find that, in general $(h, h' \in \mathcal{H})$, our model infers the expert predictions more accurately than both baselines and this competitive advantage comes from instances in which the observed prediction is by an expert $h$ who belongs to the same group of mutually similar experts as the expert $h'$ whose prediction we infer $(h, h' \in \psi)$. In fact, the GNB+CNB baseline is more accurate whenever both experts $h$ and $h'$ do not belong to the same group $(h \in \psi, h' \in \psi', \psi \neq \psi')$. Moreover, we also find that the GNB+CNB baseline infers the expert predictions more accurately whenever both experts belong to the same group of mutually similar experts identified by Algorithm 1. In Appendix 4, we report the confusion matrix of the above counterfactual predictions.

Finally, we report the per-expert $h'$ accuracy achieved by our model and both baselines in Figure 4.[15] The results show that, in general $(h, h' \in \mathcal{H})$, our model infers the expert predictions more accurately than the baselines for a majority of the experts (103 and 89, out of 114, compared to GNB and GNB+CNB, respectively). Moreover, if we restrict our attention to observed label predictions by experts

---

[15]Whenever $h, h' \in \psi$, we could not compute the per-expert accuracy for 11 experts—seven of these experts belong to singleton groups and the remaining four do not predict any of the same test samples predicted by other experts in their mutually similar groups.

$h$ belonging to the same group of mutually similar experts as the expert $h'$ whose prediction we infer $(h, h' \in \psi)$, our model infers the expert prediction more accurately for almost all experts $h'$ (100 and 101, out of 103, compared to GNB and GNB+CNB, respectively). Additionally, Figure 5 in Appendix 4 shows that, for most experts (87 out of 103), the GNB+CNB baseline infers the expert predictions $Y_{h'}$ more accurately if the observed prediction $Y_h$ is by an expert $h$ belonging to the same group of mutually similar experts as the expert $h'$ $(h, h' \in \psi)$ than if it is by an expert $h$ belonging to a different group $(h \in \psi, h' \in \psi', \psi \neq \psi')$.

# 8  CONCLUSION

In this work, we have addressed the problem of inferring second opinions by experts from the perspective of counterfactual inference. We have focused on a multiclass classification setting and showed that, if experts make predictions on their own, the underlying causal mechanism generating their predictions needs to satisfy a desirable set invariant property. Moreover, we have introduced the set invariant Gumbel-Max structural causal model, a new class of structural causal model whose structure and counterfactual predictions about second opinions by experts can be validated using interventional data.

Our work opens up many interesting avenues for future work. For example, we assume experts do not communicate before forming their opinion. Although this assumption may be satisfied in some real-world applications, it would be interesting to relax it. Moreover, we have validated our model using a single real dataset. It would be valuable to validate our model using additional datasets from other applications. Finally, it would be important to carry out user studies in which the inferred second opinions provided by our model are shared with domain experts (*e.g.*, medical doctors).

## Acknowledgements

M. Gomez-Rodriguez acknowledges support from the European Research Council (ERC) under the European Union's Horizon 2020 research and innovation programme (grant agreement No. 945719).

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
