# OpenReview forum: "Counterfactual Inference of Second Opinions"
_auai.org/UAI/2022/Conference — UAI 2022 Poster_

### Official Review · Reviewer_LC62 · 2022-04-12

**Q2(1) Originality/Novelty:** 2
**Q2(2) Significance/Impact:** 2
**Q2(3) Correctness/Technical Quality:** 2
**Q2(6) Clarity Of Writing:** 2
**Q6 Overall Score:** 5
**Q8 Confidence In Your Score:** 4

**Q1 Summary And Contributions:**

This paper focuses on the problem of inferring second opinions by experts in view of counterfactual inference. When the property of set invariant and SCM (2) holds, this article shows that the equivalence of counterfactual distribution and conditional intervention distribution.

**Q2 Assessment Of The Paper:**

More detailed information regarding each of these aspects is given below:

**Q2(5) Reproducibility:**

3: Good: Key resources (e.g., proofs, code, data) are available and key details (e.g., proofs, experimental setup) are sufficiently well-described for competent researchers to confidently reproduce the main results.

**Q3 Main Strengths:**

This paper establishes a set invariant property and two   additional desirable properties for   designing and training SI-SCM based decision support systems, which is a contribution to the literature. The integration of machine learning and counterfactual framework is a promising
approach to improve reliability of    decision support systems.

**Q4 Main Weakness:**

However, I feel that the main results and the models of this paper are ambiguous without adequate illustrations and therefore their applicability is doubtful.


**Q5 Detailed Comments To The Authors:**

1. The title of this paper seems not quite match the main results. The theoretical results of the paper is to identify counterfactual distributions based on SCM(2), and the second opinion of experts is only an   application of the proposed method.
2. In the Abstract, the authors mentioned that   “if experts make predictions on their own, the underlying causal mechanism generating their predictions needs to satisfy a desirable set invariant property.”

I have the following questions,
(i) “If experts make predictions on their own”, does this mean that each expert's data was sampled independently or   each expert does not directly affect the outcome of others?

(ii) Set invariant (Definition 1) is the only condition for Corollary 2, the conclusion of which implies that the counterfactual distribution and the conditional intervention distribution are equivalent. Combined with Section 5，how is your approach different from the following setting?

The data for each expert is an independent realization, or at least each expert in a certain group/cluster is an independent realization. Given SCM (2), it is also possible to classify/partition experts first, and then use historical data to estimate the conditional distribution for each group. We thus use the resulting model to make predictions.

3. What is the intuition of equation (4)? How does it show the stability between two experts?

4. Can you say more  about the GUMBEL-MAX SCM considered in Section 5. What are the advantages of this model? Do other commonly used SCMs for multivalued outcome satisfy the property of set invariance, such as the multinomial logistic distribution?

5. How do we sample $u_1,\ldos,u_t$ in (5)? Do we have to specify the distribution of noise variable $U$ for estimation?

Minor:
 1. The definition of interventional counterfactual SCM is not easy to understand.
  2. Theorems 2-3 have no intuitive explanations, and people may not know how to apply the results. The notation $f^\prime$ in Theorem 3 is not defined.


**Q7 Justification For Your Score:**

I am not sure about the importance of  the set invariant property  in the decision support systems the authors considered.
The lack of illustrations of results and models further  further puzzles me.

**Q9 Complying With Reviewing Instructions:**

1: Yes.

---

### Official Review · Reviewer_E56E · 2022-04-12

**Q2(1) Originality/Novelty:** 3
**Q2(2) Significance/Impact:** 2
**Q2(3) Correctness/Technical Quality:** 3
**Q2(6) Clarity Of Writing:** 3
**Q6 Overall Score:** 7
**Q8 Confidence In Your Score:** 3

**Q1 Summary And Contributions:**

The paper addresses the problem of inferring second opinions by experts from the perspective of counterfactual
inference in a  multiclass classification setting. It shows that, if experts make predictions
on their own, the underlying causal mechanism generating their predictions needs to satisfy a desirable set invariant
property. To do so, the paper define of a set invariant Gumbel-Max structural causal models.




**Q10 Ethical Concerns (Optional):**

none.

**Q2 Assessment Of The Paper:**

More detailed information regarding each of these aspects is given below:

**Q2(4) Quality Of Experiments (Optional):**

2: Fair: The experimental evaluation is weak: important baselines are missing, or the results do not adequately support the main claims.

**Q2(5) Reproducibility:**

3: Good: Key resources (e.g., proofs, code, data) are available and key details (e.g., proofs, experimental setup) are sufficiently well-described for competent researchers to confidently reproduce the main results.

**Q3 Main Strengths:**

Nice idea and well presented. Experiments  show that the model can be used to infer second opinions more accurately than its non-causal
counterpart.


**Q4 Main Weakness:**

I is a pity not having experimented both on more datasets and involving domain experts, to validate the claims.

**Q5 Detailed Comments To The Authors:**

Only few minor things:

- in SCM definition, why is the prior distribution P not a parameter explicit in the model? Also explain what is meant for "P(U)"-almost evrywhere
- end of section 2: you introduce Y variables, without saying who they are. These are explaned only later in Sect. 3, so I suggest to explain them in sect.2.

**Q7 Justification For Your Score:**

Appears to be a solid paper and well presented.

**Q9 Complying With Reviewing Instructions:**

1: Yes.

---

### Official Review · Reviewer_nskZ · 2022-04-12

**Q2(1) Originality/Novelty:** 2
**Q2(2) Significance/Impact:** 2
**Q2(3) Correctness/Technical Quality:** 3
**Q2(6) Clarity Of Writing:** 3
**Q6 Overall Score:** 5
**Q8 Confidence In Your Score:** 2

**Q1 Summary And Contributions:**

The paper proposes a pipeline allowing to train a predictor which, given the opinion of an expert on some instance, infers the opinion of other experts on the same instance. This counterfactual inference problem is equivalent to conditional inference when using an adequate "set invariant" version of a model found elsewhere, where counterfactually similar experts share a latent noise- requiring to solve a difficult combinatorial problem. An experimental study illustrates the approach.

**Q2 Assessment Of The Paper:**

More detailed information regarding each of these aspects is given below:

**Q2(4) Quality Of Experiments (Optional):**

3: Good: The experimental evaluation is adequate, and the results convincingly support the main claims.

**Q2(5) Reproducibility:**

3: Good: Key resources (e.g., proofs, code, data) are available and key details (e.g., proofs, experimental setup) are sufficiently well-described for competent researchers to confidently reproduce the main results.

**Q3 Main Strengths:**

The paper is well written, in an educational style.
The model is interesting and seems relevant.
The notion of counterfactual similarity seems deep.
The experimental results look promising.

**Q4 Main Weakness:**

The contribution seems incremental wrt the previous installment of the Gumbel-max structural causal model. Indeed, while set invariance is clearly a relevant and well discussed notion, it is technically quite straightforward to model and implement.
The idea of defining a similarity relation between experts is clever, but apparently not original, and from there the reduction to clique covering is nice, but also quite natural.
The experimental part is well executed, but focuses on a single dataset, which offers a nice proof of concept but cannot be considered conclusive wrt the practical relevance of the approach.

**Q5 Detailed Comments To The Authors:**

The paper is on the verbose side concerning counterfactual reasoning, without being precise enough: references to the 3rd level of the ladder of causation by Pearl are vague and not helpful, and the paper spends a lot of time answering questions that nobody asked in the first place (concerning counterfactual reasoning vs reasoning from data). I duly note that these questions *are* relevant, as they allow to characterize the adequate fragment of the SCM model.

**Q7 Justification For Your Score:**

The paper spends a lot of time answering questions that nobody asked in the first place. If the authors had taken a more down-to-earth approach of presenting their model, the paper would be quite blatantly too meager to warrant publication at UAI.

**Q9 Complying With Reviewing Instructions:**

1: Yes.

---

### Official Review · Reviewer_Bxk6 · 2022-04-12

**Q2(1) Originality/Novelty:** 3
**Q2(2) Significance/Impact:** 2
**Q2(3) Correctness/Technical Quality:** 3
**Q2(6) Clarity Of Writing:** 4
**Q6 Overall Score:** 7
**Q8 Confidence In Your Score:** 3

**Q1 Summary And Contributions:**

An approach is proposed to infer second opinions of experts, an important task for resource allocation. The framework builds on counterfactuals and Gumbel-Max Structural Causal Models.
Given independence between expert opinions, the authors show that 1) counterfactual distributions equal conditional interventional distributions and 2) the full problem can be partitioned into subproblems with a common noise term. Inferring these is done using clique partitioning by a random greedy algorithm.


**Q2 Assessment Of The Paper:**

More detailed information regarding each of these aspects is given below:

**Q2(4) Quality Of Experiments (Optional):**

3: Good: The experimental evaluation is adequate, and the results convincingly support the main claims.

**Q2(5) Reproducibility:**

3: Good: Key resources (e.g., proofs, code, data) are available and key details (e.g., proofs, experimental setup) are sufficiently well-described for competent researchers to confidently reproduce the main results.

**Q3 Main Strengths:**

Motivation of the work is clear (second opinions as resource allocation problem), and well illustrated by example in the medical domain.
Well written and structured, good plots. Also, intuition behind some math is given for better understanding.


**Q4 Main Weakness:**

While motivating the paper with a medical example, the evaluation is given on CIFAR-10H. Using instead a medical dataset would be more in line.

**Q5 Detailed Comments To The Authors:**

The explanation/motivation is given by medical diagnosis, but experiments with image labeling only - would be nice to also illustrate it with medical data (as Q4).

Experimental part: description of the single steps for the different approaches could be illustrated by a flow chart or similar (the paper is well written but also very dense - figures/images would help understanding).

Figure 2 in the appendix is very nice - if possible move to the main paper. Helps a lot for understanding and motivation.
Here: “consensus” given as an option to decide the final outcome based on several expert opinions - would be interesting to discuss different possibilities further.

Otherwise well written, and I really appreciate the intuition and examples given to explain the math.


**Q7 Justification For Your Score:**

Well written and structured, seems pretty solid.

However, I am not an expert on that topic.


**Q9 Complying With Reviewing Instructions:**

1: Yes.

---

### Decision · Program_Chairs · 2022-05-15

**Decision:**

Accept (Poster)

**Comment:**

Meta Review: This paper applies counterfactual inference in an original way with the goal of predicting the opinion of experts other than those used in a ML problem (for instance to define the labels in a classification problem). This has the advantage of helping with deciding whether a "second opinion" is needed and more generally to evaluate the different experts. Based on this idea, the authors analyse the features that an underlying causal model must have (in particular some invariances) depending on the relation between different experts (such as independence), and come up with a relation of all this to Gumbel-Max structural causal models.

Even though the reviewers have pointed out a few weaknesses of the paper, such as experiments (could be strengthened) and applicability (possibly unclear), I believe this paper makes nice points and applies counterfactuals in an original way that could lead to a number of offsprings in the future. More generally speaking, all reviewers lean towards acceptance (some of them weakly), and the discussion looks informative and on the point.